# The People and Their Animal Other: Representation, Mimicry and Domestication

Laurin Mackowitz

Department of Philosophy, University of Graz, 8010 Graz, Austria; laurin.mackowitz@uni-graz.at

**Abstract:** Animal stereotypes are used to describe, circumscribe and label people. They also serve to negotiate what counts as familiar and what is expelled as foreign. This article explores the composition of animal stereotypes and examines why they continue to influence the way humans understand themselves. Referring to dehumanising language in contemporary political discourse, anthropological theories of mimicry and representation as well as ethnological observations of human–animal relations, this article argues that if animals are regarded as intelligent and compassionate rather than irrational or violent, the debasing intent of animal stereotypes fails. While a deeper understanding of the mutual dependence of humans, non-humans and their environment is of academic and social interest alike, the projection of images of oneself onto animal others only highlights certain features, whilst leaving others in the dark.

**Keywords:** human–animal studies; identity formation; domestication; co-evolution; posthumanism

## 1. Introduction

In many parts of the world, including the most industrialised and urbanised areas of the Earth, animals play a significant role in human survival, well-being and development. Animals serve humans as suppliers of meat, skin, bones and other materials; as draft-animals and for other labour-intensive tasks; as experts for searching or guarding; as companions; and as testing animals for biochemical research. When referring to animals, humans relate their existence to their living environment and mirror their organic constitution, their instinctual actions and emotional intelligence. As metaphors, representations and symbols, animals provide analogies for imagination and speculation. Moreover, as rhetorical and epistemological tools, images of the animal other are fundamental sources for reflecting on social identities.

The article discusses how the representation, mimicry and domestication of animals influence the way humans (even humans who have very little physical contact with animals) understand themselves, what they are and what they are not by referring to animal stereotypes and metaphors. Human–animal relations influence the way humans define themselves: who they are, what they want and how they act. When animals are excluded with reason from the mass of the people, the very same reason is used through analogy to exclude others whose existence or actions resemble a projected image of the animal other, i.e., by being savage and irrational, without consciousness or conscience.

The distinction between animals and humans is charged with metaphysical meanings; it generates feelings, tells stories and is itself an actor. The place and manner in which a boundary is drawn between animals and humans conditions the relationship between the foreign and the own, the other and the same, as well as between nature and culture, civilisation and barbarism. On this border, human animals negotiate the bestiality of humanity and the humanity of the beast.

Examining dehumanising language in contemporary political discourse, anthropological theories of mimicry and representation as well as ethnological observations of human–animal relations, this article argues that if animals are not regarded as irrational,

savage or violent, but as intelligent, civilizational and compassionate, the debasing intent of animal stereotypes fails and returns as the celebration of the animal other as resourceful and irreplaceable.

The second section of this article questions the ways in which animal stereotypes are utilised in contemporary political discourses, exploring how they lead to an exclusionary dynamic of identity formation and the dehumanisation of political opponents. The third section discusses the depiction of animals in prehistoric art as early reflections about animals and the desires, fears, imaginations and abstractions with which images of animals are linked. The fourth section reflects on the imitation and mimicry of animals by humans, discussing the question of how human and non-human animals learn to act (appropriately and/or successfully) by imitating others. The fifth section discusses the dynamics and the consequences of domestication, as well as the close relationship between animals and humans for mutual gain. Reflecting on the significance of the animal other for co-evolution, the article argues for recognising animals not merely as dangerous beasts or friendly companions, but as active agents and partners in cultivating their common environment and social relations. Whatever characteristics human animals project onto the animal other, parts of their otherness will ultimately remain incomprehensible. The concluding section returns to the political dimension of animal stereotypes and relates the representation, imitation and domestication of animals to the ways human societies make use of the image of the animal other for manufacturing collective identities, as well as for understanding themselves and others in the mirror of their closest non-human relatives.

The article links philosophical, anthropological, political and psychoanalytic theories, namely, Jean-Yves Chateau, Paul Shepard, Rosi Braidotti, David Livingstone Smith, Hans Blumenberg, André Leroi-Gourhan, George Bataille, Claude Leví Strauss, Helmuth Plessner, Dieter Claessens, Julia Kristeva, Bernhard Waldenfels, Donna Haraway, Michel Foucault and James C. Scott. Examples from historical investigations (Peter Linebaugh and Marcus Rediker), ethnographic descriptions (Liam M. Brady, John J. Bradley, Natasha Fijn and Susan McHugh) and contemporary political discourse support the argument.

## 2. Discrimination

In his preface to Gilbert Simondon's *Two Lessons on Animal and Man*, philosopher Jean-Yves Chateau argues that everyone (including children) has an opinion about the difference between humans and animals. This distinction shapes everyday experiences both with and without animals. It influences the image one has of oneself and the expectations people have of their fellow human beings, as well as their idea of humanity. It defines "what one can hope from life" and what lies beyond it [1] (p. 11). Humans comprehend themself and their place in nature through animals, who are related and similar, and at the same time completely alien and different. Human animals recognise their constitutive other in the image of the animal other.

Animal stereotypes are used to label people as docile herbivores or aggressive predators; as submissive dogs, strong oxen, busy bees, sly foxes, royal lions, cruel wolves, filthy pigs or parasitic rats. Boundaries between the living and the dead are negotiated by referring to animal stereotypes and relationships between what is familiar and what is foreign, and are constructed around the image of animals. As metaphors and figures of thought, animals frame arguments by substituting the incomprehensible and abstract with concrete, simplifying images.

At the same time, animals are reduced to anthropomorphic stereotypes, while humans are described purely in terms of projected animal qualities. It is important to note that the moral values attributed to certain animals are subject to cultural and historical change, and can, in some aspects, even be paradoxical, as Paul Shepard demonstrates in his study *The Others: How Animals Made Us Human* with the example of the dog, which is "valorized as the companion of wandering ascetics, redeemable, welcomer of the dawn, mediator to the other world" and demonized as "the alien monster and hypocrite, fallen and hateful, the most corrupt of animals" [2] (p. 64).

The discrimination between humans and animals is of particular political importance when one considers the way the homogeneity of the collective subject of "the people" is constructed against an imagined opponent. As a supplement for the lack of a positive description of national unity, efforts to manufacture the fantasy of a homogenous community resort to identifying "the people" by excluding, negating and discriminating everything that they should not be. Evoking the difference between human and non-human animals reinforces a fictitious group identity. In this regard, philosopher Rosi Braidotti argues in *The Posthuman* that the rational self-conscious subjectivity of humanism is constructed in opposition to "sexualized, racialized, and naturalized others, who are reduced to the less than human status of disposable bodies" [3] (p. 15).

Recorded history is full of examples of discriminating humans along the lines of animal stereotypes, as the historians Marcus Rediker and Peter Linebaugh point out in *The Many Headed Hydra: Sailors, Slaves, Commoners, and the Hidden History of the Revolutionary Atlantic.* Pre-Enlightenment philosopher Francis Bacon, for example, argued for the support of colonisation and slavery in *An Advertisement Touching an Holy War* (1622) by identifying the native people of the Americas as "a 'swarm' of bees, 'shoal' of seals or whales, a 'rout' of wolves" [4] (p. 39) and as "wild and savage people [. . .] like beasts and birds" [4] (p. 61). He thus provided, as Rediker and Linebaugh argue, "a theory of genocide" [5] (p. 61). Purported animal characteristics helped to establish moral and legal categories that made it possible to dehumanise and exploit people who were not regarded as equal parts of the population.

Similar rhetoric persists today: when Donald Trump, during his presidential campaign 2023, referred to his political opponents as "thugs that live like vermin within the confines of our country that lie and steal and cheat on elections" [6], he drew on the history of symbolisation of human behaviour, virtues and vices, in the image of animals, be it the industrious ant; the stubborn donkey; or verminous cockroaches, fleas and lice.

Political animal metaphors appear to be effective instruments for stabilising collective identities, especially when facing crisis, as the following example shows. On 9 October 2023, Israel's minister of defence, Yoav Gallant, announced the complete siege of the Gaza Strip and declared: "We are fighting human animals and we act accordingly" [7]. Israel's war against Hamas was thereby framed from its very beginning with a dehumanising language that prefigured Israel's subsequent transformation of the Gaza Strip into what appears, with the same metaphorical background, as a "slaughterhouse". The statement implies a valorisation along the line of a savage-civilised trajectory that presents the war as part of civilisational efforts, aimed at domesticating what is wild, uncultivated and uncivilised, or even at domesticating what appears to be outside a purported consensus of what it means to be human. In *Making Monsters: The Uncanny Power of Dehumanization*, David Livingstone Smith argues that such dehumanising rhetoric is "demonizing" others by attributing "sinister and malevolent" characteristics onto them by labelling immoral or dangerous traits as "monstrous" [8] (p. 251).

At the same time as animal metaphors are dehumanising and delimitating, they render the incomprehensible, the fearsome and the sublime understandable and concrete. To employ an animal metaphor, metaphors "tame" uncertainties like humans tame wild animals. Calling Hamas terrorists "human animals" helps to process their cruelty by putting it in some more familiar framework and thereby pacifying the shock and terror instilled by the terrorist attack of 7 October 2023. While addressing other humans as "human animals" is dehumanising for those addressed, it is reassuring for those who assert their moral, technological and even evolutionary superiority by referring to others as animals. Referring to enemies, opponents or even uninvolved others as "human animals" is a way to legitimise extreme and violent measures against them and simultaneously stabilise collective identities under pressure by providing a very concrete negative fantasy.

Moreover, the discursive success and political efficiency of animal metaphors appear to coincide with their ability to translate an ambiguous and uncanny encounter into a more standardised image that resolves and naturalises the conceptual tension. During his

presidential election campaign in 2023, Argentine president Javier Milei declared, "I did not come here to herd sheep but to awaken lions" [9]. Identifying himself as a male lion and "king of the jungle", Milei evoked the ancient symbol of royal authority and compressed conflicting emotions, such as the admiration of strong leaders, their performance of toxic masculinity and the fear of economic and state brutality [10].

The image of the lion appears not to be a choice of Milei's personal flavour, but part of a canon of authoritarian political rhetoric. As the metaphorologist Hans Blumenberg bluntly writes in *Lions*, killing their own children, lions are cannibals and, in that sense, perform a "caricature" that is "aping humans" [11] (p. 13) (author's translation). Thus, the question of who is imitating whom not only concerns the lion and the human, but also language and politics. Are politicians merely utilising inert animal stereotypes, or do animal metaphors have their own dynamic and even agency when it comes to shaping political discourse? Considering the influence of animal stereotypes as well as their cultural contingency, the next section reflects upon the question of how and why humans began to identify themselves and others with images of animals.

## 3. Representation

The imagery of prehistoric cave paintings can be interpreted as early representations of a conscious awareness of mutual interspecies dependence and co-evolution. Many prehistoric cave paintings depict large mammals such as bison, horses, pigs, lions, bears, mammoths and deer. These images sometimes overlap with each other, and are occasionally intertwined with hybrid creatures composed of human and animal body parts.

In *Gesture and Speech*, the palaeontologist André Leroi-Gourhan argues that theories that interpret these paintings as pure descriptions of phenomena observed in nature fail to acknowledge the more complex multifunctional concept of mimesis, which combines aesthetic and symbolic functions by creating "mythograms" that prefigure script [12] (p. 202). "The representations of human beings seem to have lost all their realistic character and are now oriented toward the triangles, rectangles, and rows of lines or dots [...]." [11] (p. 191). This assemblage includes shapes and species, horns, tails, eyes, ears, arms, genitals, dots, lines and geometric shapes. These elements follow "along a trajectory and connected with one another by the link of a theme whose meaning escapes us but which is repeated again and again" [12] (p. 327) in the form of significant assemblies of "a perfectly symbolical synthetic creature" [12] (p. 395), creating a linear narration of intermingling species and their relations to each other.

This depiction of humans and animals might already be interpreted as a kind of script, "obviously [...] the picture of a vulva and a phallus scratched into a block of stone by Aurignacians is not pornography [...] but rather a [...] conception of a universe in which contrasting phenomena supplement each other." [12] (p. 395). Cave paintings depict, as Leroi-Gourhan writes, "reference systems [...] based on the alternation of opposites—day/night, heat/cold, fire/water, man/woman, and so on" [12] (p. 395).

In a related line of reasoning, philosopher George Bataille writes in *Erotism: Death and Sensuality* that prehistoric cave paintings cannot be reduced to a "magical significance" [13] (p. 74) that guarantees a successful hunt and thus reduces the cave painting to an economic utilitarian function [13] (p. 75). Instead, Bataille contends that the numerous depictions of killing and death demonstrate that they are reflections on "the ultimate in human experience" [13] (p. 75). In this sense, the animal iconography portrayed in cave paintings becomes a resource for attaining an abstract understanding of oneself and one's destiny within one's living environment.

Although more recent scholarship on the symbolic function of prehistoric cave paintings, has highlighted the limited and Eurocentric scope of Leroi-Gourhan's interpretations of prehistoric cave paintings [14] (p. 36), findings of prehistoric art outside of Europe, such as the geometric engravings on a "polished block of hematite bearing" in Blombos cave, South Africa, support his analysis of the ideographic and symbolic function of prehistoric

art, as Jean Clottes writes in *What Is Paleolithic Art?: Cave Paintings and the Dawn of Human Creativity* [14] (p. 34).

Bataille's emphasis on the reflective function of prehistoric animal representations is also supported by evidence from outside Europe. In "'That Painting Now Is Telling Us Something': Negotiating and Apprehending Contemporary Meaning in Yanyuwa Rock Art, Northern Australia", Liam M. Brady and John J. Bradley stress that the rock art of Indigenous Australians negotiates meaning by reflecting on "all the things that people know (e.g., knowledge of kin, [alive and deceased], Dreamings, spirits, country)" [15] (p. 104), whereby the position of the artists within (as well as their obligations to) their spiritual nourishing environment is reflected in animal representations [15] (p. 106).

## 4. Mimicry

Animal metaphors are well-suited tools for thinking about the obscure extremes of our existence, which we fear and label monstrous and which we aspire to and idealise. Animals are familiar and relatable and, at the same time, strange and enigmatic. As the anthropologist and ethnologist Claude Lévi-Strauss succinctly notes in *Totemism*, "natural species" are particularly important for the representation of the abstract, "not because they are 'good to eat' but because they are 'good to think'" [16] (p. 89). "The animals in totemism cease to be solely feared, admired, envied: their perceptible reality permits the embodiment of ideas and relations conceived by speculative thought on the basis of empirical observations" [16] (p. 89).

Looking at animals, humans recognise that they are mortal creatures, and by painting animals, they reflect on their mortality. Animals resemble humans in many respects, yet they are conceptualised as fundamentally different within the dominant tradition of Western thought. When describing the difficulty of grasping the paradoxical relationships between oneself and others, the psychoanalyst Julia Kristeva claims that the reflection of oneself in the other shapes our desires and fears. In *Tales of Love* she argues: "Love thus appears as an area of freedom because it accepts the protagonists' dissimilarity and even their conflict, in the same way as it aims for the identification of the one with the image of the Other—of man as an animal centered in the precedence of his needs with an ideality that is nevertheless deified because it is assumed to be accessible" [17] (p. 166). The imagination of the other in the image of the animal not only shapes how one perceives oneself and others, but also prefigures desires and fears. Images of the animalistic other are bred and cultivated "along with its retinue of idealisations and mysteries" and in "the strained motion of condensation" of contradictory or "paradoxical" signifiers [17] (p. 367), Kristeva argues.

On a related line of reasoning, the philosopher Bernhard Waldenfels writes in *Phenomenology of the Alien* that the self and the other are like two sides of a coin. One side contains all the experiences of belonging, familiarity and availability, while the other side encompasses all the ideas of non-belonging, incomprehensibility and inaccessibility [18] (p. 74). The close relationship between humans and animals enables the comprehension of the foreign and the other in the image of familiar and comprehensible, perceptible and categorizable encounters with animals. "Dogs are among Plato's favorite animals", Waldenfels writes; "he endows them with a philosophical import [sic] because they are willing to learn and capable of distinguishing the familiar from the unfamiliar, the home (οίκε̈ιον) from the alien (ἀλλότριον), and thus the friend from the foe" [18] (p. 60).

Like dogs, humans also learn who is a friend and who is not. In his anthropological account of the actor, philosopher Helmuth Plessner describes how painting and theatre create images that enable people to slip into alien roles and reflect on their eccentric positionalities, i.e., images that create a reflective distance from their environment and themselves to gain self-consciousness. "People detach themselves from themselves, transform themselves into others. They play another being." [19] (p. 404) (author's translation) When imitating the facial expressions of the other, the human actor grasps the "image-related nature of human existence" [19] (p. 417) (author's translation).

This mimicry is, as anthropologist and sociologist Dieter Claessens also writes, the characteristic for "curiosity-creatures" whose behaviour is shaped, to a considerable extent, by the imitation of the behaviour of other creatures [20] (p. 34) (author's translation). Here, the imitation of the animal and the reflection on the relationship between humans and the surrounding nature are intertwined. In a section of *The Others* titled "Aping the Others", Paul Shepard also argues that "the imitation of animals, which, however capricious, lays the groundwork for understanding ourselves as being: as actors" [2] (p. 82).

## 5. Domestication

Inquiring the ways in which human and nonhuman animals imitate each other, Natasha Fijn writes in her study of Mongolian pastoral culture *Living with Herds*, that the narrative of "wolves nurturing young "wild" children" appears in many Western and Eastern cultures [21] (p. 208). This narrative about the "reversal of the usual relationship of humans nurturing the canine" suggests that humans imitate animals, and that, vice versa "wolves have the potential to become "domestic" and humans the potential to be "wild"" [21] (p. 208). Although the stereotypes attributed to animals are in many ways anthropomorphic, Fijn's observation highlights that the mimicry of animals is not one-sided, but mutual.

In the introduction to their interdisciplinary investigation of domestication "Documenting Domestication: Bringing Together Plants, Animals, Archaeology, and Genetics" archaeologists Melinda A. Zeder, Daniel G. Bradley, Eve Emshwiller and Bruce D. Smith describe domestication as a "synergistic process" and a "coming together of a plant or animal population with a human population in an increasingly dependent mutualism" [22] (p. 2). From this perspective, domestication appears to be not a power relation, subduction or enslavement, but a "symbiosis, mutualism, or 'co-evolution' among species—with or without the involvement of humans" [23] (p. 99), as Emshwiller writes.

Despite the combination of archaeological, anthropological; and geological methods; radiocarbon and uranium–thorium dating; and the chemical analysis of organic artifacts, seeds, pollen or bones; the puzzle of how humans and animals started to cooperate remains unsolved, as does the question of why humans and animals began to cohabitate in permanent settlements and develop tools, scripture and social systems optimising this co-evolution. The improvement of archaeological methods and the discovery of new scientific facts fuel speculation about why people settled down, domesticated animals or founded cities and empires in the first place. These speculations fill the gaps between what is known and found to be true and what is unknown and not (yet) explained. The agricultural foundations of urban societies just as much form the beginning of human history as they encompass the beginning of domesticated plants and animals. Hence, the origin and the childhood of domestication also remains "among the 'big questions' of archaeological inquiry" [22] (p. 1).

Although there are common characteristics of many processes of domestication, such as selection according to tameness [21] (p. 20), Fijn stresses that domestication is not a homogeneous or ahistorical technique, and is often different from the anthropocentric theory: "a herd animal succumbs to domestication at the whims of a superior human through intentional design" [21] (p. 243). Considering the significance of the environment as well as the mutualism, cross- and interspecies dependency and "co-domestication" of Mongolian herding communities, Fijn concludes: "Herd animals, other beings, and the landscape are all actors in, and an active part of, the sociality of herding life" [21] (p. 243). In this symbiotic setting, it is impossible to decide who and what imitates, influences, feeds or depends upon whom: "herder and herd animal enculturate each other" and their environment [21] (p. 241).

The negotiation of what virtues, vices, desires or fears are attributed to animal others includes non-human actors such as animals and the environment, just as well as the cultural and technological conditions under which they are painted and composed. In this regard, biologist and feminist theorist Donna Haraway describes domestication in *The Companion*

*Species Manifesto: Dogs, People, and Significant Otherness* as all but "the paradigmatic act of masculine, single-parent, self-birthing, whereby man makes himself repetitively as he invents (creates) his tools" [24] (p. 27). Domestication is presented as the archetypical beginning of sovereign rule over others by punishment and reward. Instead, Haraway describes animals as "partners" and domestication as "an emergent process of co-habiting, involving agencies of many sorts" [24] (p. 30), agencies that are—and remain—outside the reach of self-determination.

While the representation, mimicry and domestication of animals that has been discussed above mainly referred to mammals, the co-evolution of life on earth also includes other species, such as plants and bacteria. Famously, "human gut tissue cannot develop normally without colonization by its bacterial flora", as Haraway writes [24] (p. 31). Organic life is never completely autonomous, but a heterogeneous cooperation of many interdependent species. Although all these species can be described as intertwined and, to varying degrees, interdependent, religious and scientific endeavours have differentiated them into systems of order, distinguishing and separating the chaos of life into categorizable units which are more the product of analytical rationality than an operational or pragmatic separation.

In the introductory paragraph of *The Order of Things: An archaeology of the human sciences*, philosopher Michel Foucault demonstrates that these analytic taxonomies are at times arbitrary, serving the interests of those doing the categorising more than the beings being categorized. Foucault quotes from "a certain Chinese encyclopaedia", invented by the poet Jorge Luis Borges: "animals are divided into: (a) belonging to the Emperor, (b) embalmed, (c) tame, (d) sucking pigs, (e) sirens, (f) fabulous, (g) stray dogs, (h) included in the present classification, (i) frenzied, (j) innumerable, (k) drawn with a very fine camelhair brush, (l) et cetera, (m) having just broken the water pitcher, (n) that from a long way off look like flies" [25] (p. xvi). The "oddity" of this "classification"; the paradox and "unusual juxtaposition" of general and idiosyncratic categories; and the "sudden vicinity" of political, agricultural, mythological and subjective perspectives emphasize that "the mere act of enumeration that heaps them all together has a power of enchantment" [25] (p. xvii) and that "all the ordered surfaces and all the planes with which we are accustomed to tame the wild profusion of existing things" are related to the contingency of distinguishing between the same and the other. This is a contingency that disturbs and threatens the accustomed order of things, and which demonstrates that the boundary that separates the same and the other is the product of political negotiation and never guaranteed [25] (p. xvi).

The categorisation of biological types or species is not only scientific, but also political, in as much as it is "about defining difference, rooted in polyvocal fugues of doctrines of cause", as Haraway argues [24] (p. 15). The identification of "what counts as biological kind troubles previous categories of organism" [24] (p. 15), which is why demonstrating co-evolution and challenging the fundamental measure that defines the difference between humans and animals interrupts the political order of things, its hierarchies and its discriminations.

One example of renegotiating human–animal relations is Euripides' Drama *Βάκχαι* (the Bakkhai or Bacchantes). The tragedy begins with the failure to honour Dionysus, who has returned to Thebes. In revenge, Dionysus casts a spell on all women that makes them flee the city and withdraw to the countryside to live freely and joyfully in communion with wild animals. Advocating for harmony, the chorus assesses the dispute over the merits and demerits of urban life in section 919, with a conciliatory tribute to the achievements of the hunting, herding and farming culture:

"You lead me forward, so it seems, as a bull,
You seem to have grown two horns upon your head.
Were you, all this time, an animal?
For you have certainly been . . . bullified" [26] (§ 1054–1056)

For the sake of peace, the competition between city and country should be reconciled by recognising the work of the shepherds, their dogs and their flocks for the prosperity of

the city. When the urban elite fails to appreciate the value of their periphery, the struggle threatens domesticated life with its beginning and end. Dogs, oxen and donkeys demand recognition from people, since they not only work for their labour, but carry and pull people into the harmony of tamed wilderness.

Although animal populations in urbanised and industrialised areas have diminished, and even pigeon populations have been declining due to a changing environment, as a study of the genomic diversity of passenger pigeons shows [27], the significance of non-human animals for human wellbeing persists. Even though machines have replaced animal labour in many cases, with horse-drawn carts and oxen-driven ploughs being substituted by electric or fuel-powered engines in many places, the practice of employing animal labour endures, exemplified by the ongoing training of dogs for military purposes and pigs for truffle detection. In *Seeing like a State: How Certain Schemes to Improve the Human Condition Have Failed*, anthropologist and political scientist James C. Scott asserts, in this regard, that "Agricultural engineers" replaced animals wherever possible, rationalising and standardising agriculture by promoting "mechanization" and replicating "the features of the modern factory" [28] (p. 197).

Together with a shifting importance of animals, the symbolic value of animals is changing: for example, traditionally, the ownership of animals such as camels was a signal of "prestige and political independence", but camels been rebranded as a "symbol of backwardness" for people with strong interests in industrialisation and urbanisation, as Susan McHugh argues in "Loving camels, sacrificing sheep, slaughtering gazelles: human–animal relations in contemporary desert fiction" [29] (p. 180). The symbolic value attributed to animals is intricately tied to natural and political landscapes, as well as the cultural and economic environment in which they exist and are constructed. Conversely, the conceptualisation of environments and societies is linked to the ways in which animals are imagined. Early humans painted and performed the wild animals they feared and hunted; with the invention of domestication, they castrated aurochs and harnessed them to ploughs. As a result, labour power multiplied, and with labour power came surplus value. In turn, neolithic humans learned to subject themselves to burdens, yokes and harnesses, loading both themselves and their animals with weights and oppression in ways, which led Scott to ask ironically in *Against the Grain: A Deep History of the Earliest States*: "And what about the "domesticators in chief", Homo sapiens? Were not they domesticated in turn, strapped to the round of ploughing, planting, weeding, reaping, threshing, grinding, all on behalf of their favorite grains and tending to the daily needs of their livestock? It is almost a metaphysical question who is the servant of whom—at least until it comes time to eat" [30] (p. 32).

## 6. Conclusions

The persistent presence of animal metaphors in contemporary political discourse indicates that animals continue to be enduring figures in the collective imagination of urbanised and industrialised societies. Even though the values and attributes linked with certain wild or domesticated animals are changing due to the transformation of landscapes, economies, lifestyles and ideologies, the phantasmatic figures of kind and cruel animal others continue to haunt and animate city-dwellers. These figurations resonate with the lived experiences of semi-nomadic pastoralists whose lives depend on the accurate assessment of the risks and benefits posed by wild or domestic animals. While the use of animal metaphors may be divisive, the ambiguity of animal stereotypes, their anthropomorphic familiarity and their projected foreignness are simultaneously blurring any attempt to arrive at a clear definition of what is meant or intended when others are referred to as vermin, lions or human animals. By obfuscating zoological standardisations, animal stereotypes compress imagined fears and historical and fashionable representations with real and fictious imitations of animals. Moreover, the images attributed to animals are open to change, which is why their effects depend on the way they are composed and performed—if the projected animal other instils terror and justifies violence or if it motivates compassion, humility and cooperation.

Human animals represent, imitate and domesticate images of the animal other to legitimise and stabilise political communities. They cultivate animal stereotypes to understand themselves and their mortality as part of and in mutual exchange with their living environment as well as their communication with nonhuman animals. Emphasising the significance of animals and recognising their real differences while observing their similarities may also change the anthropocentric image of the animal other. Respecting their undisguised otherness and sameness has the potential to transform the debasing discourses that unify social groups against each other and against their environments.

Although the values associated with animals are subject to transformation, cultivating and studying animal metaphors implies the recognition of their own dynamic, their genealogy and their agency. The study of animal stereotypes may collect and catalogue hegemonial, outdated or peripheral meanings associated with animals. And it may distinguish the subtle connotations and evocations implicit in animal imaginaries. An in-depth understanding of the mutual dependence between humans, non-humans and their environment is of academic and social interest alike. The projection of emotions or characteristics (which may be only accessible or evident to oneself) onto others, be they lifeforms or non-organic in nature, shines a light onto certain features whilst leaving others in the dark. Although images of animal others are useful for epistemic and political reasons, one should be cautious and aware that, although these projections suggest similarities, some features of the other will always remain obscure and incomprehensible.

**Funding:** This research was funded by the State of Styria [UFO2022PN29].

**Institutional Review Board Statement:** Not applicable.

**Informed Consent Statement:** Not applicable.

**Data Availability Statement:** The study did not report any data.

**Conflicts of Interest:** The author declares no conflict of interest.

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
