# Peer review of "The People and Their Animal Other: Representation, Mimicry and Domestication"

_philosophies, doi:10.3390/philosophies9010003_

Round 1

Reviewer 1 Report

Comments and Suggestions for Authors

While this is a well-intentioned paper with a coherent message, it would benefit from some significant revision. The argument is clear however the uncritical use of sources lets it down. Many of these theories have been subject to very valid criticisms which really need to be incorporated into the paper. There is also a glaring omission in the writings of Paul Shepard which speak directly to this subject matter. The arguments also suffer from a Western bias which reflects many of the sources.

20-23: This is not the case for millions of people in developing countries who still use animals for transport and cultivation.

31: Animals also provide some people with prestige

113 - Those cave paintings are actually late representations of consciousness, specific to a European prehistoric culture and separate from other cultures which had developed in many other parts of the world by that stage. Cave paintings seem early because they survive erosion and weathering while other, older forms of art have not. They do not speak to the origins of human thought about animals but represent human engagement with animals in a articular context.

119 - Leroi-Gourhan's work has been subject to criticism - western bias, speculation, over simplification. These should be recognised

143 - No serious archaeologist would agree that prehistoric cave paintings mark the birth of art. Bataille's prioritising Lascaux here verges on racism

172-173 - This is a very Western view based on a distinction between farm animal and pet to which many cultures do not subscribe.

200-215 - The claims here are based on selectivity on what is human, wolf, and chimp behaviour. These ethologists have generalised based on very narrow experiments. Humans in no way owe their social skills to wolves. Homo sapiens only encountered wolves in the last 100k years and in places such as Australia have lived without wolves for >70k years. Again there is a western bias here.

209 - The history of 'man' is no longer acceptable terminology

211 - 'We' is a very generalised term. There are many people who protest or boycott circuses and zoos

214 - I may be misunderstanding this but I doubt people work in slaughterhouses in order to be close to animals

p. 218 - here is a narrow, western view of civilisation that ignores Indigenous civilisations. It is also misguided to conflate "scientific, technological, economic, moral and evolutionary progress" as these operate independently and certainly do not define civilisation

227 - this would benefit from the work of Natasha Fijn

243 - This claim has no evidence to support it, see above comments on humans and wolves

254 - The conclusion should not have new information added. While it should not be a place holder, it should summarise the paper without introducing new ideas and evidence.

300-314 - This seems to be the argument of the paper. In which case it should be stated clearly, early on in the paper, and the paper should be structured and written to support the argument and identify any weaknesses. It is a worthy argument that deserves better support. I would suggest reading ethnographic material on humans and animals in a more global context.

Comments on the Quality of English Language

The English is overall excellent with only a few corrections needed

Author Response

For research article

Response to Reviewer 1 Comments

1. Summary

2. Questions for General Evaluation

Reviewer’s Evaluation

Response and Revisions

Does the introduction provide sufficient background and include all relevant references?

Must be improved

I thoroughly revised the introduction and also introduced the references, I am drawing on.

Are all the cited references relevant to the research?

Yes

Are the research design, questions, hypotheses and methods clearly stated?

Can be improved

I have rewritten the introduction and clarified my hypothesis

Are the arguments and discussion of findings coherent, balanced and compelling?

Must be improved

I have incorporated new research supporting my argument.

For empirical research, are the results clearly presented?

Is the article adequately referenced?

Not applicable

Yes

Are the conclusions thoroughly supported by the results presented in the article or referenced in secondary literature?

Must be improved

I have completely rewritten my conclusions.

3. Point-by-point response to Comments and Suggestions for Authors

Comments 1:

While this is a well-intentioned paper with a coherent message, it would benefit from some significant revision. The argument is clear however the uncritical use of sources lets it down. Many of these theories have been subject to very valid criticisms which really need to be incorporated into the paper. There is also a glaring omission in the writings of Paul Shepard which speak directly to this subject matter. The arguments also suffer from a Western bias which reflects many of the sources.

Response 1: Thank you for pointing at the limits and bias of some of my sources. I agree with this comment. Therefore, I have deleted some of the more problematic statements and included supporting literature for others [line 333 - line 347]. Thank you for mentioning Paul Shepards work. I have included two of his arguments [line 219-216 and line 395-397].  

Comments 2:

20-23: This is not the case for millions of people in developing countries who still use animals for transport and cultivation.

Response 2:

Agree. I deleted the passage in question, and I have included ethnographic descriptions of human-animal relations in developing countries [line 420 –427, line 450 – 460, and line 554 –559].

Comments 3:

31: Animals also provide some people with prestige

Response 3:

Agree. I have included reflections on how animals provide prestige and provided reference for how this prestige has been changing over time [line 554 –559].

Comments 4:

113 - Those cave paintings are actually late representations of consciousness, specific to a European prehistoric culture and separate from other cultures which had developed in many other parts of the world by that stage. Cave paintings seem early because they survive erosion and weathering while other, older forms of art have not. They do not speak to the origins of human thought about animals but represent human engagement with animals in a particular context.

Response 4:

Agree. I have deleted the passage in question and have included reference to South African prehistoric art instead [line 333 - 339].

Comments 5:

119 - Leroi-Gourhan's work has been subject to criticism - western bias, speculation, over simplification. These should be recognized.

Response 5:

Agree. I have included more recent scholarship on prehistoric art that problematizes but also supports Leroi-Gourhan’s emphasis on the symbolic function of prehistoric art [line 333 - 339].

Comments 6:

143 - No serious archaeologist would agree that prehistoric cave paintings mark the birth of art. Bataille's prioritising Lascaux here verges on racism

Response 6:

Agree. I have deleted the passage in question.

Comments 7:

172-173 - This is a very Western view based on a distinction between farm animal and pet to which many cultures do not subscribe.

Response 7:

Agree, this is very biased, and I have deleted the passage in question.

Comments 8:

200-215 - The claims here are based on selectivity on what is human, wolf, and chimp behaviour. These ethologists have generalised based on very narrow experiments. Humans in no way owe their social skills to wolves. Homo sapiens only encountered wolves in the last 100k years and in places such as Australia have lived without wolves for >70k years. Again there is a western bias here.

Response 8:

Agree, the evidence is sparce and I have deleted the passages in question.

Comments 9:

209 - The history of 'man' is no longer acceptable terminology

Response 9:

Agree, the terminology is outdated and I have deleted the passage in question, since it was not essential for my argument.

Comments 10:

211 - 'We' is a very generalised term. There are many people who protest or boycott circuses and zoos.

Response 10:

Agree, I have deleted the passage in question.

Comments 11:

214 - I may be misunderstanding this but I doubt people work in slaughterhouses in order to be close to animals.

Response 11:

Agree, I have deleted the passage in question.

Comments 12:

p. 218 - here is a narrow, western view of civilisation that ignores Indigenous civilisations. It is also misguided to conflate "scientific, technological, economic, moral and evolutionary progress" as these operate independently and certainly do not define civilization.

Response 12:

Agree, I thought to include reference to Hans Peter Duerr’s critique of the “myth of civilization”, in the end I have deleted the whole passage, since it was not relevant for the argument.

Comments 13:

227 - this would benefit from the work of Natasha Fijn

Response 13:

Thank you very much for mentioning Fijn’s research. I have included arguments from her book Living with Herds about cultural differences and similarities as well as her argument for the interdependence of co-domestic species and their environment [line 420-427 and 450-460].

Comments 14:

243 - This claim has no evidence to support it, see above comments on humans and wolves

Response 14:

Agree, I have deleted the passage in question.

Comments 15:

254 - The conclusion should not have new information added. While it should not be a place holder, it should summarise the paper without introducing new ideas and evidence.

Response 15:

Agree, I have rewritten the whole conclusion and summarized the argument.

Comments 16:

300-314 - This seems to be the argument of the paper. In which case it should be stated clearly, early on in the paper, and the paper should be structured and written to support the argument and identify any weaknesses. It is a worthy argument that deserves better support. I would suggest reading ethnographic material on humans and animals in a more global context.

Response 16:

Agree, I have rewritten the introduction and the conclusions. I have also included ethnographic material as well as reference to contemporary political discourse to support the philosophical discussion.

Comments on the Quality of English Language

The English is overall excellent with only a few corrections needed

Response:

Thank you for pointing at the quality of my English. Nevertheless, I have corrected some grammar mistakes in collaboration with a native English speaker.

Reviewer 2 Report

Comments and Suggestions for Authors

An important topic that has been expanded and described well with sound arguments. 

Author Response

For research article

Response to Reviewer 2 Comments

1. Summary

2. Questions for General Evaluation

Reviewer’s Evaluation

Response and Revisions

Does the introduction provide sufficient background and include all relevant references?

Yes

Are all the cited references relevant to the research?

Yes

Are the research design, questions, hypotheses and methods clearly stated?

Not applicable

Are the arguments and discussion of findings coherent, balanced and compelling?

Not applicable

For empirical research, are the results clearly presented?

Is the article adequately referenced?

Not applicable

Yes

Are the conclusions thoroughly supported by the results presented in the article or referenced in secondary literature?

Yes

3. Point-by-point response to Comments and Suggestions for Authors

Comments 1:

An important topic that has been expanded and described well with sound arguments.

Response 1: Thank you for recognizing the importance of the article.

Reviewer 3 Report

Comments and Suggestions for Authors

This essay purports to demonstrate the continued usefulness of animal others for human self-identity. However, in several places in the article, the author restates their purpose with a new twist, so ultimately it is difficult to identify the true thesis. For example, at line 64, the author states that they want to argue that animals are allies, not enemies. At line 107, the author states that they wish to unpack animal stereotypes and their use in manufacturing collective identities. These themes may be interrelated, but nonetheless, the overall purpose of this essay remains murky.

While I applaud the general topic of this essay as one that is important for our times, I do not see anything in this essay that is new or interesting. The role of animal others in human history and in the myriad ways we interact with them, use them – for their bodies, labor power, as symbols or as mirrors – has been widely and deeply studied in many disciplines. Sociologists, geographers, anthropologists, biologists, ethicists, and historians have elaborated on just about every aspect of this topic we can imagine in an extremely rich and rigorous literature, none of which is cited here. Unfortunately, this essay uses a very limited set of sources, all of which are very old. The most recent is 2017 and most are much older. 

The essay hints at some lines of investigation that could be fruitful, such as the use of animal images and stereotypes in emerging political conflicts in the present moment. But this topic is only suggested here, not explored.

The essay is weak in other important ways. There are many grammatical errors and some of the writing is very confusing; a deep edit for standard English is needed. For example, the subject of the sentence at line 30 seems to be “animals” but this makes the meaning completely opaque.

At line 59, the author introduces notions of populism and “corrupt elites” and “criminal migrants.” This reviewer has no idea what the author is referring to here.

The paper uses a number of very long quotes which don’t seem to help the narrative.

There are also factual errors in the way the author represents the evolution of human-animal relationships. The opening lines of the essay call into question what part of the world the author is focusing on, for human-animal relations have not evolved in the same way everywhere. For example, use of animals for their labor in the fields is still quite common around the world, outside of the Global North. Additionally, while it is true that people in post-domestic societies such as our own rarely encounter animals except as pets, that does not mean that we are unconnected to a multitude of animals that are raised in factory farms for our benefit or provide fodder for science and medicine, or hides for our belts and shoes. A much deeper understanding of the human connection to animal others is required.

I would suggest the author step back and take some time to read the most recent scholarship on non-human animals, their role and meaning in our lives in the Anthropocene, and then start again to try to develop a new contribution to this literature.

Comments on the Quality of English Language

Please see comments above. A deep edit is needed here for grammar and structure.

Author Response

For research article

Response to Reviewer 3 Comments

1. Summary

2. Questions for General Evaluation

Reviewer’s Evaluation

Response and Revisions

Does the introduction provide sufficient background and include all relevant references?

Must be improved

I thoroughly revised the introduction and also introduced the references, I am drawing on.

Are all the cited references relevant to the research?

Can be improved

I have included recent research and deleted less relevant references.

Are the research design, questions, hypotheses and methods clearly stated?

Must be improved

I have rewritten the introduction and clarified my hypothesis.

Are the arguments and discussion of findings coherent, balanced and compelling?

Must be improved

I have incorporated new research supporting my argument.

For empirical research, are the results clearly presented?

Is the article adequately referenced?

Not applicable

Must be improved

I have included recent research as well as historical and contemporary examples, supporting my argument.

Are the conclusions thoroughly supported by the results presented in the article or referenced in secondary literature?

Must be improved

I have completely rewritten my conclusions.

3. Point-by-point response to Comments and Suggestions for Authors

Comments 1:

This essay purports to demonstrate the continued usefulness of animal others for human self-identity. However, in several places in the article, the author restates their purpose with a new twist, so ultimately it is difficult to identify the true thesis. For example, at line 64, the author states that they want to argue that animals are allies, not enemies. At line 107, the author states that they wish to unpack animal stereotypes and their use in manufacturing collective identities. These themes may be interrelated, but nonetheless, the overall purpose of this essay remains murky.

Response 1: Thank you for pointing out that the focus of my paper is not clear. I have rewritten the introduction to clarify that the aim of the paper is to question animal stereotypes and their usage for manufacturing collective identities. 

Comments 2:

While I applaud the general topic of this essay as one that is important for our times, I do not see anything in this essay that is new or interesting. The role of animal others in human history and in the myriad ways we interact with them, use them – for their bodies, labor power, as symbols or as mirrors – has been widely and deeply studied in many disciplines. Sociologists, geographers, anthropologists, biologists, ethicists, and historians have elaborated on just about every aspect of this topic we can imagine in an extremely rich and rigorous literature, none of which is cited here. Unfortunately, this essay uses a very limited set of sources, all of which are very old. The most recent is 2017 and most are much older.

Response 2:

I agree that the topic has been explored before. I have included more recent literature on the subject. I have discussed contemporary political discourse, which has not been discussed in this format before. I have linked philosophical and anthropological theories with contemporary, historical and ethnographic examples in a novel way and thereby contributed to the discussion of human-animal relations, the projection of human characteristics onto animals and vice versa as well as to research on the manufacturing, justification and stabilization of collective identities.

Comments 3:

The essay hints at some lines of investigation that could be fruitful, such as the use of animal images and stereotypes in emerging political conflicts in the present moment. But this topic is only suggested here, not explored.

Response 3:

Agree. I have expanded this topic in a whole new section that discusses the use of animal stereotypes in current political debates [line 204-296].

Comments 4:

The essay is weak in other important ways. There are many grammatical errors and some of the writing is very confusing; a deep edit for standard English is needed. For example, the subject of the sentence at line 30 seems to be “animals” but this makes the meaning completely opaque.

Response 4:

Agree. I have edited the whole paper together with a native English speaker, correcting many grammatical errors and clarifying the argument.

Comments 5:

The paper uses a number of very long quotes which don’t seem to help the narrative.

Response 5:

Agree. I have deleted some direct quotes and paraphrased others.

Comments 6:

There are also factual errors in the way the author represents the evolution of human-animal relationships. The opening lines of the essay call into question what part of the world the author is focusing on, for human-animal relations have not evolved in the same way everywhere. For example, use of animals for their labor in the fields is still quite common around the world, outside of the Global North. Additionally, while it is true that people in post-domestic societies such as our own rarely encounter animals except as pets, that does not mean that we are unconnected to a multitude of animals that are raised in factory farms for our benefit or provide fodder for science and medicine, or hides for our belts and shoes. A much deeper understanding of the human connection to animal others is required.

Response 6:

Agree. I have included more recent ethnographic literature on human-animal relations in pastoral as well as urban communities by Liam M. Brady and John J. Bradley, Natasha Fijn, and McHugh to support my argument. And I have discussed the cultural dependence of animal stereotypes.

Comments 7:

I would suggest the author step back and take some time to read the most recent scholarship on non-human animals, their role and meaning in our lives in the Anthropocene, and then start again to try to develop a new contribution to this literature.

Response 7:

Disagree, I have systematically revised my article and included more recent research on human-animal relations and the construction of animal stereotypes, which is why I think that it is an important and original contribution to the subject.

Comments on the Quality of English Language

Please see comments above. A deep edit is needed here for grammar and structure.

Response:

I have corrected many grammar mistakes in collaboration with a native English speaker.

Round 2

Reviewer 1 Report

Comments and Suggestions for Authors

The changes have really improved the article. I am impressed with the improvements given the short period of time

Comments on the Quality of English Language

The article needs editing again as there are a few errors.

Author Response

Response to Reviewer 1

Second Round

1. Summary

2. Questions for General Evaluation

Reviewer’s Evaluation

Response and Revisions

Is the content succinctly described and contextualized with respect to previous and present theoretical background and empirical research (if applicable) on the topic?

Yes

Are all the cited references relevant to the research?

Yes

Are the research design, questions, hypotheses and methods clearly stated?

Not applicable

Are the arguments and discussion of findings coherent, balanced and compelling?

Yes

For empirical research, are the results clearly presented?

Is the article adequately referenced?

Not applicable

Yes

Are the conclusions thoroughly supported by the results presented in the article or referenced in secondary literature?

Yes

.

3. Point-by-point response to Comments and Suggestions for Authors

Comments 1:

The changes have really improved the article. I am impressed with the improvements given the short period of time

Response 1: Thank you for your comment. I have revised the article again and clarified my argument further.  

Comments on the Quality of English Language

The article needs editing again as there are a few errors.

Response:

I have revised the article in collaboration with a native English speaker and corrected grammatical errors and clarified many sentences.

Reviewer 3 Report

Comments and Suggestions for Authors

I am impressed with the changes you have made to this manuscript - great job on that. I think it reads much more coherently, is much better embedded in some of the literature, and more accurate in its discussion of the human-animal relationship. There are a few places where I am not sure how the discussion supports the overall goal of the paper - for example, the section on Euripides seems a bit out of place. There are still sections that need more clarity in the writing. The second sentence of the first paragraph of the conclusion is a run-on sentence - very confusing.

My major concern is that the focus on the use of animal stereotypes in contemporary politics which is now clearly emphasized a part  of the thesis, is still somewhat underdeveloped. It would be great to include some additional examples if possible - understanding that word limit is always a challenge. Also, one has to wonder if we did adopt a more universally compassionate and respectful attitude toward animals, would animal metaphors disappear entirely, or would they shift? Some animal metaphors are laudatory. "Cunning as a fox," can be a compliment. "Strong as an ox," is also a positive statement. It just isn't clear to me what you think should happen. Your conclusion section could use more work - it is sort of a "kitchen sink" approach and isn't very specific about what you want to see change.

Comments on the Quality of English Language

This is an improvement but still needs some work. One sentence I identified in the first draft is unchanged and I still have no idea what it means.

Author Response

Response to Reviewer 3

Second Round

1. Summary

2. Questions for General Evaluation

Reviewer’s Evaluation

Response and Revisions

Is the content succinctly described and contextualized with respect to previous and present theoretical background and empirical research (if applicable) on the topic?

Yes

Are all the cited references relevant to the research?

Yes

Are the research design, questions, hypotheses and methods clearly stated?

Can be improved

I have added a statement in the conclusion, specifying the research framework.

Are the arguments and discussion of findings coherent, balanced and compelling?

Can be improved

I have clarified some sentences.

For empirical research, are the results clearly presented?

Is the article adequately referenced?

Not applicable

Yes

Are the conclusions thoroughly supported by the results presented in the article or referenced in secondary literature?

Can be improved

I have revised my conclusions and added some clarifications.

3. Point-by-point response to Comments and Suggestions for Authors

Comments 1:

I am impressed with the changes you have made to this manuscript - great job on that. I think it reads much more coherently, is much better embedded in some of the literature, and more accurate in its discussion of the human-animal relationship. There are a few places where I am not sure how the discussion supports the overall goal of the paper - for example, the section on Euripides seems a bit out of place. There are still sections that need more clarity in the writing. The second sentence of the first paragraph of the conclusion is a run-on sentence - very confusing.

Response 1: Thank you for your comment. I introduced the Euripides example and provided some context for why I am mentioning it here. I have also clarified many sentences, including the one pointed out above.

Comments 2:

My major concern is that the focus on the use of animal stereotypes in contemporary politics which is now clearly emphasized a part of the thesis, is still somewhat underdeveloped. It would be great to include some additional examples if possible - understanding that word limit is always a challenge. Also, one has to wonder if we did adopt a more universally compassionate and respectful attitude toward animals, would animal metaphors disappear entirely, or would they shift? Some animal metaphors are laudatory. "Cunning as a fox," can be a compliment. "Strong as an ox," is also a positive statement. It just isn't clear to me what you think should happen. Your conclusion section could use more work - it is sort of a "kitchen sink" approach and isn't very specific about what you want to see change.

Response 2: Thank you for your comment. I could not include another example for contemporary political usage of animal metaphors. There was no space for it. But I have clarified the passage about transforming animal stereotypes and emphasized that the positive and negative values attributed to animals are changing. In the conclusion I also suggested how research on animal metaphors and stereotypes can contribute to a better understanding of the variety of hegemonial and obscure meanings attributed to animals.

Comments on the Quality of English Language

This is an improvement but still needs some work. One sentence I identified in the first draft is unchanged and I still have no idea what it means.

Response:

I have revised the article in collaboration with a native English speaker and corrected grammatical errors and clarified many sentences.
